# Hawler Medical University students' perceptions of e-learning during the COVID-19 pandemic

**Sherzad A. Shabu** *

Department of Community Medicine, College of Medicine, Hawler Medical University, Erbil, Iraq

* sherzad.hakim@hmu.edu.krd

## Abstract

**Data Availability Statement:** All relevant data are within the paper and its Supporting Information files.

**Funding:** The Author received no specific fund for this work.

### Background

Protective measures like social distancing and staying home when possible were imposed by the public health authorities to combat the COVID-19 pandemic. Educational institutions which had previously experienced only face-to-face traditional methods of service delivery suffered from huge difficulties in maintaining an effective teaching process. This cross-sectional study investigates the experiences of students of Hawler Medical University (Erbil, Iraqi Kurdistan), exploring their e-learning experience and satisfaction when conventional classroom learning was suspended due to lockdown.

### Methods

A self-administered online questionnaire designed on Google Forms was administered from 14 to 31 March 2020. The main section of the questionnaire asked participants to rate their agreement with statements concerning e-learning satisfaction using a five-point Likert scale, with answers ranging from *strongly disagree* to *strongly agree*.

### Results

From an initial 1550 participants, 808 ultimately completed the processual stage. The majority of respondents were female, from the College of Medicine, of urban residence, and of an average economic situation. Three-quarters of the respondents had not participated in any online course before, 27.7% did not have the required electronic devices, 43.3% did not have the sufficient computer skills for the e-learning, and 38% thought that the University did not support the use of the e-learning system. During the processual stage, only 24.4% thought that the internet connection was good and helpful, 14.6% easily attended the online sessions, 35.6% thought that teachers have enough time to answer students' questions and concerns, and 18% that the teaching materials contributed to the course objectives and overall learning outcome.

**Competing interests:** The Author has declared that no competing interests exist.

## Conclusions

Hawler Medical students were not well prepared to engage in the e-learning process. The majority experienced internet access and other technical problems in accessing the online sessions. They did not feel that the e-learning process has contributed to the achievement of course objectives and overall learning outcomes.

## Introduction

From late 2019 COVID-19 rapidly spread worldwide, and from March 2020 governments worldwide applied compulsory lockdowns or social distancing recommendations that resulted in most educational institutions closing or restricting traditional campus and on-site learning processes, causing an abrupt shift to ad hoc online solutions, to enable some degree of continuity [1, 2]. Iraq including the Kurdistan Region, has also experienced an increasing number of confirmed COVID-19 cases during the first week of April 2020. The Kurdistan Regional Government (KRG) tried to implement some strict measures to contain the COVID-19 outbreak, like suspending all the schools and universities in the region [3].

Just before the pandemic, the need for distance learning has just begun to develop as a novel mode of education, with some tentative and experimental applications, but the rupture caused to traditional educational formats due to lockdowns entailed a hasty and unprepared shift to the wholesale provision of online learning as the only way of education and lecturing for most people [4].

Worldwide, different academic institutions have experienced the application of e-learning or distance learning for many years, but many of them had not implemented such learning as a standalone package before the emergence of COVID-19 pandemic. While a limited number of academic institutions with more robust experience of implementing such programs could easily repurpose and deploy off-the-shelf packages during the COVID emergency, the majority of institutions worldwide were wrong-footed, being used to only face-to-face interaction program experience. They consequently suffered from huge difficulties in maintaining an effective teaching process, and the prolonged duration of lockdown in educational institutions in most countries began to have increasingly serious long-term academic, educational, and economic implications for individual students and national economies [4, 5].

Therefore, and in order to compensate the obvious disruption of the education process, higher education authorities were required to issue new rules and regulations, mainly in terms of adopting the e-learning system and making use of advanced technologies to continue the education process and mitigate the impact of COVID-19 pandemic lockdowns [5]. In all countries, the use of e-learning systems might be a challenge for many universities in itself, but this is likely to be a more acute concern in developing countries, especially in relation to students' willingness to adopt and accommodate the e-learning system. This is due to increased barriers to the accessibility of required technologies for e-learning systems in developing countries [6]. Also, the lack of the required knowledge in information technology and poor networking infrastructure could be additional challenges to accepting and effectively using such systems [7]. Financial constraints have been identified as the most fundamental obstacle to applying e-learning systems in most contexts [8].

Considering that this is a novel and dynamic research issue *per se*, and that there are limited studies concerning Iraqi Kurdistan to determine how students are prepared to engage in e-learning and how they perceive it in terms of its strengths and weaknesses, this study

investigates Hawler Medical University (HMU) students' preparedness to engage in e-learning and explores their perceptions of its advantages and disadvantages.

## Methods

### Design and setting

Google Form was used in this cross-sectional study to administer a self-completed online survey of students at HMU (Erbil, Iraqi Kurdistan) from 14 to 31 March 2020 (during the first lockdown of educational institutions in the region). HMU was established in July 2005, and includes five colleges (Medicine, Dentistry, Pharmacy, Nursing, and Health Sciences). The University has an average student body of 3600 students, who were considered to comprise the study population. All students of the five colleges of HMU were invited to participate in this survey.

Social networking sites like Viber, WhatsApp, and Twitter were used to share the survey link with students, in addition to the official university's e-mail addresses. The invitation message and the introduction part of the online survey explained the purpose of the survey to the participants, requested a consent to participate, and explained the anonymity of the participants and confidentiality of the collected information.

The study was carried out in two stages: (1) the initial stage before the start of the e-learning process to explore students' preparedness and their feeling towards the start of such process; and (2) the processual stage during the implementation of e-learning to explore students' perception about the all process, including its strengths and weaknesses.

### Survey tool

The survey's questionnaire for the both stages was designed to get a rapid appraisal of the HMU students' perception and attitude towards e-learning. The questionnaire was adapted from a study conducted to develop a questionnaire for predicting online learning achievement [9]. The initial stage questionnaire included three main sections, concerning participants': (1) socio-demographic characteristics; (2) number of online courses taken previously, and number of hours per week using computers for educational activities etc.; and (3) a five-point Likert-type list of statements concerning their e-learning experience, which they were asked to rate (strongly disagree to strongly agree).

The latter (i.e., Likert) questions were mainly about whether students: have the required electronic devices for the e-learning; are able to easily access the internet for that; feel comfortable communicating electronically; understand the study topics; have sufficient computer skills required for the e-learning; achieve sufficient interaction with their teachers; find learning outside the class more motivating than on-campus; consider that the University supports the use of e-learning; and require an induction session about e-learning to familiarize them with the process, etc.

For the processual stage questionnaire, the same form of questions based on a Likert scale were followed, and these included questions relating to: the quality of internet connection; the instructions provided to access the e-materials and connect to the e-sessions; whether they experienced technical problems; the variety of teaching methods used to maximize the students' learning; conducting quizzes and assignments to better understand the topics; whether the time set for the online sessions was fair; and if teachers have enough time to answer most of students' questions and concerns.

For each stage, and prior to data collection, ten Hawler Medical University students were initially invited to pilot test the questionnaire. They completed the online survey questionnaire

and tested the clarity of the questions' contents and the practicality of the questionnaire's completion. Based on the comments from these students, the questionnaire was adapted.

## Ethical considerations

This study was approved by the HMU Research Ethics Committee. Online written consent was obtained from the participants before completing the survey. All participants were informed about the voluntary nature of participation, their right to withdraw at any time prior to submitting the online form (at which point their data was anonymized and became irretrievable), and the anonymity and confidentiality of all data.

## Data analysis

SPSS (v. 22) was used for data entry and analysis. A descriptive analysis was used to calculate the frequency and proportion of the students chosen different Likert scale items for each individual question and also calculating its mean score out of five points. Chi square test of statistical significance was used to examine the association between different variables. A *p*-value equal or less than 0.05 was considered as statistically significant.

## Results

Before starting the e-learning process (initial stage), the demographic characteristics section (the results of which are summarized in Table 1) revealed that 66.4% of the total number of students (1550) were females, compared to 67% of 808 students of the processual stage (during the e-learning process). During the initial stage, the majority of students were from the College of Medicine (43.6%), compared to 39.2% during the processual stage. The majority of students were from the first and second grades, with 26.1% and 25.7% for the first grade during the initial and processual stages, and 24% and 26.2% for the second grade during the initial and processual stages (respectively). The majority of students were of urban residence (78.2% for the initial stage, and 72.6% for the processual stage). More than 68% of students were from inside Erbil City during the initial stage compared to 65.3% during the processual stage.

During the initial stage, 75.5% of the students stated that they have not taken any online courses for any reason before. A large minority (40.3%) stated that they use computers between 1–5 hours per week for educational activities, while 30.5% stated that they only use computers less than an hour; approximately 30% used computers for 6 hours or more. In terms of general internet usage, over a third spend over 10 hours online per week, while a fifth spend less than an hour or 6–10 hours, and over a quarter used it for 1–5 hours (Table 2).

Table 3 shows the Likert scale results for the initial assessment stage. More than 27% of the students did not have the required electronic devices for the e-learning process, almost one quarter of them were not able to easily access the internet, 46% were not feeling comfortable to communicate electronically, and only 26.5% were willing to communicate with their classmates and instructors electronically. Less than a quarter of them felt that they had beneficial background and experience related to e-learning, and e-learning was considered to be useful to manage study time more effectively by over 28%, and to enable them to understand study topics more easily by 21.8%.

Over 45% stated that they enjoy working independently, and 28.7% stated that they like a lot of interaction with their instructors. More than 43% of them did not have sufficient computer skills required for the e-learning, 39% did not feel comfortable communicating online in English, and almost 49% did not feel that they would be able to ask questions and get quick responses during the e-learning process.

**Table 1. Characteristics of the study participants.**

| Characteristic | Initial stage (*n* = 1550) | | Processual stage (*n* = 808) | |
|---|---|---|---|---|
| | Frequency | Percent | Frequency | Percent |
| **Gender** | | | | |
| Male | 521 | (33.6) | 267 | (33.0) |
| Female | 1029 | (66.4) | 541 | (67.0) |
| **College** | | | | |
| Medicine | 676 | (43.6) | 317 | (39.2) |
| Dentistry | 123 | (7.9) | 98 | (12.1) |
| Pharmacy | 311 | (20.1) | 192 | (23.8) |
| Nursing | 280 | (18.1) | 159 | (19.7) |
| Health Sciences | 160 | (10.3) | 42 | (5.2) |
| **Grade** | | | | |
| First | 405 | (26.1) | 208 | (25.7) |
| Second | 371 | (24) | 212 | (26.2) |
| Third | 357 | (23) | 185 | (22.9) |
| Fourth | 261 | (16.9) | 89 | (11.0) |
| Fifth | 117 | (7.5) | 66 | (8.2) |
| Sixth | 39 | (2.5) | 48 | (5.9) |
| **Residence1** | | | | |
| Urban | 1212 | (78.2) | 587 | (72.6) |
| Semi-urban | 231 | (14.9) | 161 | (19.9) |
| Rural | 107 | (6.9) | 60 | (7.4) |
| **Residence2** | | | | |
| Inside Erbil City | 1059 | (68.3) | 528 | (65.3) |
| Outside Erbil City | 332 | (21.4) | 198 | (24.5) |
| Outside Erbil Gov. | 159 | (10.3) | 82 | (10.1) |
| **Economic level** | | | | |
| Below average | 166 | (10.7) | / | / |
| Average | 1278 | (82.5) | / | / |
| Above average | 106 | (6.8) | / | / |
| **Total** | **1550** | **(100.0)** | **808** | **(100.0)** |

More than 53% of them felt that face-to-face contact with the instructor is necessary for learning, 39.7% did not feel motivated by the materials in an e-learning/online activity outside of class, 49.4% felt unable to discuss with other students during the e-learning, 52.8% felt unable to still work in a group during the e-learning, and 65.3% did not agree that learning is the same in class and at home on the internet.

Almost 62% of the students did not believe that the e-learning is more motivating than on campus, 57.6% did not believe that a complete course can be delivered by e-learning without difficulty, and 43% of them did not feel that they could pass the e-learning course with minimum teacher's assistance. Less than a quarter of the students stated that the University supported the use of e-learning, and 45.5% stated that they need to take an induction session about e-learning to be familiar with the process.

Table 4 shows the Likert scale results for the processual assessment stage. Only 24.4% of the students agreed that their internet connection was good and helpful for the e-learning process, 19.8% agreed that the instruction provided to access the e-materials and connect to the e-sessions were clear and helpful, 14.6% agreed that attending the online sessions was easy, with

**Table 2. Online activities by students (*n* = 1550).**

| Characteristic | Frequency | Percent |
|---|---|---|
| **Number of online courses taken previously (for any reason)** | | |
| None | 1171 | (75.5) |
| 1–2 | 272 | (17.5) |
| 3–4 | 56 | (3.6) |
| ≥ 5 | 51 | (6.8) |
| **Hours per week of computer use for educational activities** | | |
| < 1 | 473 | (30.5) |
| 1–5 | 625 | (40.3) |
| 6–10 | 224 | (14.5) |
| > 10 | 228 | (14.7) |
| **Hours spent online per week** | | |
| < 1 | 322 | (20.8) |
| 1–5 | 396 | (25.5) |
| 6–10 | 312 | (20.1) |
| > 10 | 520 | (33.5) |
| Total | 1550 | (100.0) |

**Table 3. Initial stage/statements about e-learning (*n* = 1550).**

| Statement | Responses *n* (%) | | | | | Mean |
|---|---|---|---|---|---|---|
| | SD | D | N | A | SA | |
| **Key** *SD*: *Strongly Disagree*, *D*: *Agree*, *N*: *Neutral*, *A*: *Agree*, *SA*: *Strongly Agree* | | | | | | |
| 1. I have the required electronic devices for e-learning | 208 (13.4) | 222 (14.3) | 402 (25.9) | 516 (33.3) | 202 (13) | 3.18 |
| 2. I am able to easily access the Internet for e-learning | 232 (15) | 307 (19.8) | 399 (25.7) | 446 (28.8) | 166 (10.7) | 2.46 |
| 3. I am comfortable communicating electronically | 311 (20.1) | 401 (25.9) | 431 (27.8) | 291 (18.8) | 116 (7.5) | 2.67 |
| 4. I am willing to communicate with my classmates and instructors electronically | 284 (18.3) | 373 (24.1) | 482 (31.1) | 320 (20.6) | 91 (5.9) | 2.71 |
| 5. I feel that my background and experience will be beneficial to e-learning | 273 (17.6) | 399 (25.7) | 506 (32.6) | 292 (18.8) | 80 (5.2) | 2.68 |
| 6. I feel that I will be able to manage my study time effectively and easily throughout e-learning | 312 (20.1) | 373 (24.1) | 430 (27.7) | 338 (21.8) | 97 (6.3) | 2.7 |
| 7. I feel that I will be able to understand the study topics easily throughout e-learning | 356 (23) | 401 (25.9) | 456 (29.4) | 252 (16.3) | 85 (5.5) | 2.55 |
| 8. As a student, I enjoy working independently | 143 (9.2) | 223 (14.4) | 480 (31) | 497 (32.1) | 207 (13.4) | 3.25 |
| 9. I like a lot of interaction with my instructors | 147 (9.5) | 244 (15.7) | 714 (46.1) | 361 (23.3) | 84 (5.4) | 2.99 |
| 10. I have sufficient computer skills for doing e-learning | 287 (18.5) | 384 (24.8) | 425 (27.4) | 339 (21.9) | 115 (7.4) | 2.74 |
| 11. I feel comfortable communicating online in 12. English | 271 (17.5) | 333 (21.5) | 404 (26.1) | 394 (25.4) | 148 (9.5) | 2.88 |
| 13. I will be able to ask my teacher questions and receive quick responses during e-learning | 330 (21.3) | 428 (27.6) | 458 (29.5) | 276 (17.8) | 58 (3.7) | 2.55 |
| 14. I feel that face-to-face contact with my instructor is necessary to learn | 175 (11.3) | 224 (14.5) | 320 (20.6) | 423 (27.3) | 408 (26.3) | 3.42 |
| 15. I am motivated by the materials in an e-learning/online activity outside of class | 221 (14.3) | 394 (25.4) | 636 (41) | 251 (16.2) | 48 (3.1) | 2.68 |
| 16. I will be able to discuss with other students during e-learning/online activities outside the class | 295 (19) | 471 (30.4) | 457 (29.5) | 283 (18.3) | 44 (2.8) | 2.55 |
| 17. I will be able to still work in a group during e-learning/online activities outside of class | 317 (20.5) | 501 (32.3) | 412 (26.6) | 272 (17.5) | 48 (3.1) | 2.50 |
| 18. Learning is the same in class (on campus) and at home on the Internet | 546 (35.2) | 467 (30.1) | 279 (18) | 191 (12.3) | 67 (4.3) | 2.21 |
| 19. I believe that e-learning outside of class is more motivating than a regular course (on campus) | 474 (30.6) | 483 (31.2) | 355 (22.9) | 157 (10.1) | 81 (5.2) | 2.28 |
| 20. I believe a complete course can be given by e-learning without difficulty | 417 (26.9) | 476 (30.7) | 351 (22.6) | 236 (15.2) | 70 (4.5) | 2.39 |
| 21. I feel I could pass an e-learning course with minimum teacher assistance | 288 (18.6) | 362 (23.4) | 495 (31.9) | 328 (21.2) | 77 (5) | 2.70 |
| 22. In general, the University has supported the use of the e-learning system | 266 (17.2) | 322 (20.8) | 589 (38) | 330 (21.3) | 43 (2.8) | 2.71 |
| 23. I need to take an induction session about e-learning to be familiar with the process | 152 (9.8) | 218 (14.1) | 474 (30.6) | 523 (33.7) | 183 (11.8) | 2.23 |

Table 4. Processual stage/statements about e-learning (*n* = 808).

| Statement | Responses *n* (%) | | | | | Mean |
|---|---|---|---|---|---|---|
| | SD | D | N | A | SA | |
| **Key** *SD*: *Strongly Disagree, D: Agree, N: Neutral, A: Agree, SA: Strongly Agree* | | | | | | |
| 1. In general, the internet connection was good and helpful | 223 (27.6) | 205 (25.4) | 183 (22.6) | 165 (20.4) | 32 (4) | 2.47 |
| 2. The instructions provided to access e-materials and connect to e-sessions were clear and helpful | 232 (28.7) | 236 (29.2) | 180 (22.3) | 130 (16.1) | 30 (3.7) | 2.36 |
| 3. Attending the online sessions was easy, with minimum technical problems | 278 (34.4) | 266 (32.9) | 146 (18.1) | 95 (11.8) | 23 (2.8) | 2.15 |
| 4. The e-links provided for the teaching materials worked properly and were easily accessible | 276 (34.2) | 248 (30.7) | 146 (18.1) | 103 (12.7) | 35 (4.3) | 2.22 |
| 5. The teaching materials used were clear/ understandable | 233 (28.8) | 222 (27.5) | 192 (23.8) | 123 (15.2) | 38 (4.7) | 2.39 |
| 6. The teaching materials used contributed to the course objectives and overall learning outcome | 219 (27.1) | 218 (27) | 225 (27.8) | 115 (14.2) | 31 (3.8) | 2.40 |
| 7. Enough and proper references were provided for the topics | 205 (25.4) | 216 (26.7) | 197 (24.4) | 155 (19.2) | 35 (4.3) | 2.50 |
| 8. Different teaching activities were used which helped us to maximize our learning | 270 (33.4) | 257 (31.8) | 173 (21.4) | 85 (10.5) | 23 (2.8) | 2.17 |
| 9. The teaching activities encouraged me to engage more and exchange ideas with other students | 303 (37.5) | 261 (32.3) | 157 (19.4) | 70 (8.7) | 17 (2.1) | 2.05 |
| 10. The quizzes/assignments helped me to better understand the topics | 342 (42.3) | 208 (25.7) | 135 (16.7) | 89 (11) | 34 (4.2) | 2.09 |
| 11. The time duration set for the online sessions was fair and sufficient | 262 (32.4) | 196 (24.3) | 179 (22.2) | 122 (15) | 49 (96.1) | 2.38 |
| 12. Teachers had enough time to answer most of our questions and concerns | 190 (23.5) | 147 (18.2) | 183 (22.6) | 178 (22) | 110 (13.6) | 1.95 |

minimum technical problems, and 17% of them agreed that the e-links provided for the teaching materials worked properly.

With regard to the teaching materials, 19.9% of them agreed that they were clear and understandable, 18% agreed they contributed to the course objectives and the overall learning outcome. Only 23.5% of the students agreed that that enough references were provided for the topics, while 13.3% considered that different (varied) teaching activities were used to maximize the learning, and only 10.8% the teaching activities encouraged them to engage more and exchange ideas with other students. Only around 15% of the students agreed that the quizzes/assignments helped them to better understand the topics, 21.1% thought the time duration set for the online sessions was sufficient, and 35.6% agreed that teachers had enough time to answer most of their questions and concerns.

## Discussion

The purpose of this study is to investigate HMU students' perceptions about e-learning during the lockdown period of COVID-19 pandemic, considering their preparedness to engage in the e-learning process and how do they perceived it during its application. Student satisfaction remains the most important and the key factor in finding out the strengths and challenges in front of applying digital learning, as with any learning solution. The results of this study are intended contribute to improving the overall e-learning process and maximize its outcome for the students.

There is an increasing need to enhance the ways in which educational institutions can develop continuously updated and improved curriculums with the most effective methods to meet learners' needs. Academic institutions are required to think about the incorporation of innovative methods of teaching into their teaching program [10]. While tentative (or token) efforts have been made worldwide to develop and incorporate e-learning solutions, the sudden compulsion to adopt e-learning as the sole platform for educational service delivery during COVID-19 lockdowns revealed the sorry state of preparedness in most institutions. Globally, the vast majority of higher education institutions were not prepared to implement the e-learning system and apply its tools as a sole method; they lacked sound platforms for digital learning, teaching staff were not equipped for practicing remote teaching, and familiarity with

online teaching was mainly confined to sharing slides, handbooks, and assignments with students via email [11].

In this study, 1550 HMU students completed the online questionnaire during the initial stage, and 808 students during the processual stage. These students were from all the five colleges of HMU, representing different grades/stages and socio-demographic characteristics (including different genders, economic backgrounds, and residential areas).

Although about one-third of the studied sample stated that they spent more than 10 hours per week online for learning purposes, almost three-quarters of them had never attended any online courses for any reason before. This could be attributed to teaching culture in our region, which is still a traditional face-to-face one despite the advent of computer technology and the internet. These results were consistent with those of a study conducted in Iran to explore the university students' viewpoints on e-learning during the COVID-19 pandemic, which revealed that 87.7% of the participants had no former experience of e-learning [12]. A study conducted on university students of Dhaka, Bangladesh, also revealed that 68.2% of students had no pre-pandemic familiarity with e-learning [13].

While only 46.3% of the students had the required electronic devices for the e-learning, only 39.5% of them could easily access the internet, and only 24.4% of them stated that during the online sessions the internet was good and helpful. Almost two-thirds of them stated that attending the online sessions was not easy, and that they encountered some major technical problems. In addition, just 29.3% of them had sufficient computer skills required to engage in the e-learning. This showed the insufficient preparedness of the students to engage in the process of digital learning from the technical point of view. The results were consistent with those of a study conducted in Jordan, which revealed that around half of students on average had prerequisite high-quality digital tools to engage in e-learning, 42% of them faced some technical problems in the online lectures due to internet connection issues, and around two-thirds of them have encountered technical problems while submitting exams electronically [14]. The results were also consistent with a qualitative study conducted in Europe to assess the challenges of e-learning during Covid-19 from sport university students perspectives, which revealed that not everyone has the same access to online teaching since students predominantly work from informal spaces not mainly prepared for learning, such as their dorms or shared flats, and usually, from their bedrooms. Based on that, students expressed their wish for a proper technological suitable environment provided by the university for a safe and quiet e-learning [15].

Only 17% of the students in this study agreed that they could easily access the teaching materials through the provided link, 19.9% agreed that the teaching materials used were clear and understandable, and just 13.3% of them agreed that different teaching activities were used to help in maximizing their learning. This could be due to the fact that neither the teaching staff were well trained, in advance, on how to upload the learning materials during the online learning, nor were the students trained on how to access them properly. A study conducted on a number of universities in India revealed that only 10.8% of the students agreed that proper learning materials were uploaded during the online learning, and that only 17.4% of them could have access to the electronic study materials [16]. Conversely, many studies emphasized the importance of distance learning in enabling students to have more resources, and the ability to reuse resources (e.g., re-watching instructional videos) [17–19].

The study showed that only 21.1% of students agreed that the time duration set for the online sessions was fair and sufficient, 10.8% agreed that the teaching activities encouraged them to engage more and exchange ideas with other students, and 35.6% of them felt that teachers had enough time to answer most student questions and concerns. These results were consistent with those of studies conducted in China, France, and South Korea, which

consistently cited a lack of interaction as one of the foremost disadvantages associated with the e-learning [20–22]. Also, in a study conducted in India, only 33.2% of students agreed that the e-learning has improved collaboration and interactivity among students [16].

Only 18% of the students in this study greed that the teaching materials used in the e-learning process have contributed to achieving course objectives and overall learning outcomes. These results are inconsistent with those of a study conducted in Jordan, which reported that 65% and 25% (Arts/Humanities and Sciences respectively) of the students agreed that the course objectives and learning outcomes were achieved through distance learning with the same degree of effectiveness as in face-to-face education [14]. Furthermore, a study conducted in India also showed that 61.4% of the students stated that they access study resources effectively and that 68.5% of them get updated learning materials through the e-learning [16].

In this study, only 24.1% of the students agreed that the University has supported the use of the e-learning system. This could be attributed to the low level of experience of university administration in engagement in such a relatively new process of e-learning, in addition to possible financial constraints. In a study conducted in Iran to find out the students' view points on e-learning, some participants considered the lack of effective cooperation and communication between groups of managers in pursuing the virtual problems of students as an important challenge in the field of e-learning process [12]. Contrary to that, 58% of the respondents of a study conducted in Ukraine to explore distance learning during COVID-19 pandemic were satisfied with the support received from the university administration for proper engagement in the process of e-learning [23].

As illustrated in Table 4 of the processual stage assessment, the overall mean calculated for each statement did not reach half of the total weight (out of 5) except for one statement: "enough and proper references were provided for the topics". This result reflects the relatively low rate of student satisfaction with the overall e-learning process. A study conducted in Bangladesh to explore university students' acceptance of e-learning during COVID-19 revealed that 29.1% of the students agreed that they are satisfied with the e-learning experience during COVID-19 pandemic [13]. The results of the study conducted in Ukraine revealed a huge difference with those of this study, which reported that the student satisfaction rates with various forms of distance learning were 66.9% (satisfied) and 18.1% (very satisfied), and only 7.9% were unsatisfied [23].

Because of the study design and mainly the use of Likert scale for data collection and organization, it was somehow difficult to consider comparison between different participant groups. Further studies which consider comparison between these groups are recommended.

## Conclusions

Hawler Medical students were not well prepared to engage in e-learning, since the majority of them had not been enrolled in any similar courses before. The majority also experienced internet access problems and other technical barriers to accessing the online sessions, hence they encountered difficulties in accessing the teaching materials through the provided links. The duration of sessions was considered insufficient, and it did not allow for adequate and fair interactions with teachers in order to answer students' concerns. Overall, students had a low satisfaction rate with the e-learning process, and they thought that it did not contribute to the achievement of their course objectives and overall learning outcomes. Therefore, necessary measures need to be adopted by the university to address all these gaps through better planning and setting clear policies to promote the e-learning practice. Managers and decision makers need to invest in developing the required structure and work on the necessary pedagogical approaches in order to improve the students' satisfaction rate with the e-learning process.

Also, further studies need to be planned and conducted in the future to better explore the existing gaps in the e-learning process. These studies could address the comparison between different groups of the students.

## Supporting information

**S1 Appendix.**
(SAV)

**S2 Appendix.**
(SAV)

## Acknowledgments

The author would like to thank Dr. Dara Al-Banna for his help in putting the survey questionnaire in "Google Form". His efforts are very much appreciated.

## Author Contributions

**Conceptualization:** Sherzad A. Shabu.

**Data curation:** Sherzad A. Shabu.

**Formal analysis:** Sherzad A. Shabu.

**Investigation:** Sherzad A. Shabu.

**Methodology:** Sherzad A. Shabu.

**Project administration:** Sherzad A. Shabu.

**Software:** Sherzad A. Shabu.

**Writing – original draft:** Sherzad A. Shabu.

**Writing – review & editing:** Sherzad A. Shabu.

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
