## [Decision Letter · Decision Letter 0]

2 Jan 2023

PONE-D-22-26736Hawler Medical University students’ perceptions of e-learning during the COVID-19 pandemic.PLOS ONE

Dear Dr. Shabu,

Thank you for submitting your manuscript to PLOS ONE. After careful consideration, we feel that it has merit but does not fully meet PLOS ONE’s publication criteria as it currently stands. Therefore, we invite you to submit a revised version of the manuscript that addresses the points raised during the review process.

ACADEMIC EDITOR:

Its a nice study. Please revise the manuscript and mention reference/s for the questionnaire.

We look forward to receiving your revised manuscript.

Kind regards,

Mukhtiar Baig, Ph.D.

Academic Editor

PLOS ONE

Journal Requirements:

" ext-link-type="uri" xlink:type="simple">https://journals.plos.org/plosone/s/file?id=ba62/PLOSOne_formatting_sample_title_authors_affiliations.pdf"

2. Please provide additional details regarding participant consent. In the ethics statement in the Methods and online submission information, please ensure that you have specified what type you obtained (for instance, written or verbal, and if verbal, how it was documented and witnessed). If your study included minors, state whether you obtained consent from parents or guardians. If the need for consent was waived by the ethics committee, please include this information

Additional Editor Comments (if provided):

Please revise the manuscript. It has been mentioned in the methodology that "The questionnaire was

117 developed based on reviewing the literature of similar studies." Therefore, please provide the references for the questionnaire.

Reviewers' comments:

Reviewer's Responses to Questions

**Comments to the Author**

1. Is the manuscript technically sound, and do the data support the conclusions?

Reviewer #1: Yes

Reviewer #2: Yes

2. Has the statistical analysis been performed appropriately and rigorously? 

Reviewer #1: I Don't Know

Reviewer #2: Yes

3. Have the authors made all data underlying the findings in their manuscript fully available?

Reviewer #1: Yes

Reviewer #2: Yes

4. Is the manuscript presented in an intelligible fashion and written in standard English?

Reviewer #1: Yes

Reviewer #2: Yes

5. Review Comments to the Author

Reviewer #1: Good effort on a burning issue. The flow of writing is fine and ideas are presented in a well structured manner. Problem and knowledge gap are identified. Results are discussed well but they are not compare enough with other studies and not critically analyze enough. I suggest if the provide further recommendations based on the conclusion of study.

Reviewer #2: Overall the study is good considering the sample size and the rigor.

However, in the methodology section the development of the tool especially with respect to the selection of the items needs more emphasis. The section says that the items were taken from literature from similar studies, but it is not clear whether the items were adapted from any similar survey questionnaire. Furthermore, whether the tool was piloted has not been mentioned. In the analysis it would have been beneficial if the authors would have considered the comparison between different participant groups.

6. PLOS authors have the option to publish the peer review history of their article (what does this mean?). If published, this will include your full peer review and any attached files.

Reviewer #1: **Yes: **Syeda Rubaba Azim

Reviewer #2: No

---

## [Author Response · Author response to Decision Letter 0]

12 Jan 2023

Dear Editor,

Thank you very much for sharing the Editor's and reviewers' comments and suggestions on my manuscript entitled " Hawler Medical University students’ perceptions of e-learning during the COVID-19 pandemic." I very much thank them for their valuable comments and suggestions. These comments have significantly helped me to improve the quality and clarity of the manuscript.

I have made the necessary revision by responding to the suggested comments. Please find below explanations to the revision made through a point-to-point response to the comments. 

Thank you very much for considering the revised manuscript.

Best regards,

Sherzad Shabu

Journal Requirements:

Author's response

PLOS ONE style requirements are applied to the manuscript.

2. Please provide additional details regarding participant consent. In the ethics statement in the Methods and online submission information, please ensure that you have specified what type you obtained (for instance, written or verbal, and if verbal, how it was documented and witnessed). If your study included minors, state whether you obtained consent from parents or guardians. If the need for consent was waived by the ethics committee, please include this information

Author's response

Online written consent was obtained from the participants before completing the survey.

Author's response

It was checked and it is complete and correct. 

Additional Editor comment

Please revise the manuscript. It has been mentioned in the methodology that "The questionnaire was 117 developed based on reviewing the literature of similar studies." Therefore, please provide the references for the questionnaire.

Author's response

The reference for the questionnaire was provided (Reference number 9). The statement was also revised under the Methods section/Survey tool, Page 6, Line number 117 and 118. 

Review Comments to the Author 

Reviewer #1: 

Reviewer’s comment 

Good effort on a burning issue. The flow of writing is fine and ideas are presented in a well structured manner. Problem and knowledge gap are identified. Results are discussed well but they are not compare enough with other studies and not critically analyze enough. I suggest if the provide further recommendations based on the conclusion of study.

Author's response

Thank you very much for the valuable comments.

- Further comparisons of results were made under the Discussion section (Page 19, Line 277-284) and (Page 20-21, Line 314-323).

- Critical analysis of results was made under the Discussion section (Page 18, Line 254-256), (Page 19, Line 288-290) and (Page 21, Line 325-327).

- Further recommendations were provided based on the conclusion of the study under the Conclusions section (Page 22, Line 363-370). 

Reviewer #2: 

Reviewer’s comment 

However, in the methodology section the development of the tool especially with respect to the selection of the items needs more emphasis. The section says that the items were taken from literature from similar studies, but it is not clear whether the items were adapted from any similar survey questionnaire. Furthermore, whether the tool was piloted has not been mentioned. In the analysis it would have been beneficial if the authors would have considered the comparison between different participant groups.

Author's response

Thank you very much for the useful comments. 

- The questionnaire was adapted from a study conducted to develop a questionnaire for predicting online learning achievement (Reference number 9). The statement was added to the Methods section/Survey tool (Page 6, Line 117-118). 

- Piloting the survey tool: A paragraph was added in Methods section/Survey tool (Page 7, Line 141-145) as follows:

For each stage, and prior to data collection, ten Hawler Medical University students were initially invited to pilot test the questionnaire. They completed the online survey questionnaire and tested the clarity of the questions' content and the practicality of the questionnaire's completion. Based on the comments from these students, the questionnaire was adapted. 

- Comparison between different participant groups in the analysis: A "Study limitations" section was added at the end of the Discussion section (Page 22, Line 350-353) stating the following:

Study limitations

Because of the study design and mainly using the Likert scale for data collection and organization, it was somehow difficult to consider comparison between different participants' groups. Further studies which consider comparison between these groups are recommended.

---

## [Editor Report · Decision Letter 1]

16 Jan 2023

Hawler Medical University students’ perceptions of e-learning during the COVID-19 pandemic

PONE-D-22-26736R1

Dear Dr. Shabu,

We’re pleased to inform you that your manuscript has been judged scientifically suitable for publication and will be formally accepted for publication once it meets all outstanding technical requirements.

Kind regards,

Mukhtiar Baig, Ph.D.

Academic Editor

PLOS ONE
---

## [Editor Report · Acceptance letter]

14 Feb 2023

PONE-D-22-26736R1 

Hawler Medical University students’ perceptions of e-learning during the COVID-19 pandemic 

Dear Dr. Shabu:

I'm pleased to inform you that your manuscript has been deemed suitable for publication in PLOS ONE. Congratulations! Your manuscript is now with our production department. 

Kind regards, 

on behalf of

Professor Mukhtiar Baig 

Academic Editor

PLOS ONE